# Generation and Transcriptome Profiling of Slr1-d7 and Slr1-d8 Mutant Lines with a New Semi-Dominant Dwarf Allele of *SLR1* Using the CRISPR/Cas9 System in Rice

**DOI:** 10.3390/ijms21155492

**Published:** 2020-07-31

**Authors:** Yu Jin Jung, Jong Hee Kim, Hyo Ju Lee, Dong Hyun Kim, Jihyeon Yu, Sangsu Bae, Yong-Gu Cho, Kwon Kyoo Kang

**Affiliations:** 1Division of Horticultural Biotechnology, Hankyong National University, Anseong 17579, Korea; yuyu1216@hknu.ac.kr (Y.J.J.); gllmon@naver.com (J.H.K.); ju950114@naver.com (H.J.L.); skullmask@naver.com (D.H.K.); 2Institute of Genetic Engineering, Hankyong National University, Anseong 17579, Korea; 3Department of Chemistry, Hanyang University, Seoul 04763, Korea; muner8146@gmail.com (J.Y.); sangsubae@hanyang.ac.kr (S.B.); 4Department of Crop Science, Chungbuk National University, Cheongju 28644, Korea; ygcho@chungbuk.ac.kr

**Keywords:** CRISPR/Cas9, GA, DELLA/TVHYNP, Dwarf, *GA_20_OX_2_*, GA signaling

## Abstract

The rice *SLR1* gene encodes the DELLA protein (protein with DELLA amino acid motif), and a loss-of-function mutation is dwarfed by inhibiting plant growth. We generate slr1-d mutants with a semi-dominant dwarf phenotype to target mutations of the DELLA/TVHYNP domain using CRISPR/Cas9 genome editing in rice. Sixteen genetic edited lines out of 31 transgenic plants were generated. Deep sequencing results showed that the mutants had six different mutation types at the target site of the TVHYNP domain of the *SLR1* gene. The homo-edited plants selected individuals without DNA (T-DNA) transcribed by segregation in the T1 generation. The slr1-d7 and slr1-d8 plants caused a gibberellin (GA)-insensitive dwarf phenotype with shrunken leaves and shortened internodes. A genome-wide gene expression analysis by RNA-seq indicated that the expression levels of two GA-related genes, *GA_20_OX_2_* (Gibberellin oxidase) and *GA_3_OX_2_*, were increased in the edited mutant plants, suggesting that *GA_20_OX_2_* acts as a convert of GA_12_ signaling. These mutant plants are required by altering GA responses, at least partially by a defect in the phytohormone signaling system process and prevented cell elongation. The new mutants, namely, the slr1-d7 and slr1-d8 lines, are valuable semi-dominant dwarf alleles with potential application value for molecule breeding using the CRISPR/Cas9 system in rice.

## 1. Introduction

Rice, being one of the major food crops consumed by nearly half of the world’s population, is grown annually at about 4.5 million hectares. Rice consumption per capita is particularly high in Asia, where it provides 60–70% calories per day (Food and Agriculture Organization (FAO), 2004). Therefore, among the breeding program priorities of rice breeders is to improve its tolerance to abiotic stress, such as tolerance to lodging. To date, the properties of many dwarf mutants found in plants have been associated with genes on the biosynthesis and signaling pathways of gibberellin (GA), indole-3-acetic acid (IAA), brassinolide (BR), and other hormones [1]. GA is one of the important plant hormones acting as a group of diterpenoid compounds that regulate during various growth and development processes in the higher plants, including stem elongation, germination, dormancy, flowering, flower development, leaf, and fruit aging [1,2,3]. The phenotypes of mutants deficient in GA biosynthesis or signaling usually exhibit dark green and rough leaves in rice [4]. So far, several genes related to defective mutants on the GA biosynthetic pathway, namely, *d18*, *d35*, *sd1*, and *eui,* have been isolated and characterized in rice [5,6,7,8]. Molecular genetic studies of GA-sensitive rice and Arabidopsis mutants have identified important factors for GA signaling, which seems to be well conserved among flowering plants [9,10]. The most important regulator of the GA signaling pathway is the DELLA protein, which is known as the repressor of GA action [11,12]. The DELLA proteins belong to the GRAS family as a transcription factor and are known to contain the N-terminal DELLA/TVHYNP amino acid motif and the C-terminal GRAS domain [13,14]. In addition, the genes encoding the GA receptor GA-INSENSITIVE DWARF1 (GID1), the F-box protein GA-INSENSITIVE DWARF2 (GID2) and the DELLA protein have been cloned, and an integrated GA signal transduction pathway has emerged [15,16,17]. Furthermore, it has been reported that DELLA family proteins interact with growth-related transcription factors such as PIF (phytochrome interaction factor) to control plant cell and organ size [18]. In general, GID1–GA–DELLA complexes in plant cells recognize GA by receptors. However, in the case of rice, the F-box protein of GID2 additionally interacts with the DELLA protein, which is polyubiquitinated by E3 ubiquitin-ligase (GID2) and then degraded through the 26S proteasome [19]. It is known that internode elongation is facilitated by GA signaling through GID1 and the DELLA protein in rice [20]. To date, accumulating evidence highlighted the N-terminus of DELLA as necessary for the inhibition of GA action. It has also recently been shown that DELLA N-terminus is required to interact with the GA receptor GID1 and consequent degradation [21]. In rice, the slr1-d1, -d2, -d3, -d4, -d5 and-d6 mutants in the GA signal transduction inhibitor DELLA protein N-terminal region consequently result in a dominant, semi-dwarf phenotype [22]. These mutants are known to have an amino acid modified by one bp substitution in the conserved DELLA/TVHYNP domain of the DELLA protein [22].

The CRISPR/Cas9 system, a recently developed genome modification tool, has been widely used for genome editing of several major crops due to its high accuracy and efficiency [23,24]. Furthermore, CRISPR/Cas9 has not only been used to knock out target genes in cells but also to introduce fragments of a certain size into the gene [25,26].

In this study, the CRISPR/Cas9 system was employed for targeting the TVHYNP domain of the *OsSLR1* gene, known as the DELLA protein. A total of six homozygous edited plants with new different allelic variants, namely, slr1-d7, slr1-d8, slr1-d9, slr1-d10, slr1-d11, and slr1-d12, showed dwarfism. In addition, mutants slr1-d7 and slr1-d8 were further investigated at transcriptome levels using RNA-sequencing.

## 2. Results

### 2.1. Editing of the TVHYNP Domain Encoding the OsSLR1 Gene and the CRISPR/Cas9 System

According to the structure of the *OsSLR1* gene, DELLA and TVHYNP domains are well conserved at the N-terminus (Figure 1A, Appendix A). Sixteen mutants were identified by single guide RNA (sgRNA) region which targeted the *OsSLR1* gene in the positive transgenic T_0_ plants (Appendix A). Deep sequence analyses detected 6 homozygous mutations, 2 heterozygous mutations, and 8 bi-allelic mutations (Appendix A). All the T_0_ mutants were dwarf, producing many tillers. Six homozygous mutants were identified, among which four were characterized by few bp deletions and two by a few bp insertions. Specifically, the following were observed: a 3-bp deletion and mutant named slr1-d7, a 1-bp deletion that was designated as slr1-d8, a 5-bp deletion named slr1d9 and a 14-bp deletion called slr1-d10. The insertion mutants were a T-insertion named slr1d11 and a C-insertion named slr1-d12 (Figure 1B). Theoretically, the slr-d7 mutant encodes a protein without serine (Ser, S) and the slr1-d8~slr1-d12 are knockout mutants with a stop codon that cannot encode the protein (Figure 1C). A single-base deletion and insertion are predicted to cause a frameshift, resulting in the knockout of the *OsSLR1* gene. However, all the mutations did not affect the core sequence TVHYNP domain of the *OsSLR1* gene. The sgRNA was also investigated using Cas-OFFinder (http://www.rgenome.net/cas-offinder/) [27], and two potential off-target sites were chosen. Interestingly, no mutations were detected in these loci. These mutations are either untranslated or modified SLR1 proteins and have different mutant sites compared to the previously reported slr1-d1~slr1-d6 allele [28,29,30].

### 2.2. New Allelic Slr-d7~Slr-d8 Mutant Plants Showed Dwarfing

The new dwarf mutants (slr1-d7~slr1-d12) showed several deficiencies in addition to reducing plant size (Appendix A). Compared to the wild type (WT), these mutants had a slow growth rate, showed dwarfed and shriveled leaves (Figure 2A). The stomata are the key channels that regulate gas exchange and water evaporation in the leaves. As a result of observing stomata sizes by SEM (scanning electron microscopy) images, the slr1-d7 and slr1-d8 lines were smaller than that of WT (Figure 2B). To observe cytological differences in the stem internodes of these mutants, paraffin sections of the stem internodes were investigated from two mutants (slr1-d7 and slr1-d8) and WT. These dwarf mutants showed that the cell size was significantly reduced, and the internode thickened as the cell layer was increased (Figure 2C). In addition, the length of all internodes of the slr1-d7 and slr1-d8 lines were reduced compared to WT (Figure 2D). These results are similar to the characteristics of *dn*-type rice dwarf mutants previously reported by Takeda [31]. Thus, slr1-d7 and slr1-d8 lines were semi-dominant dwarf mutants, indicating that a decrease in cell length may be a direct cause of shortened culm length in dwarf mutant plants. Furthermore, to know the cause of dwarfism, the length of the leaf sheath was measured according to GA_3_ concentration treatment in slr-d7, slr-d8, and WT. The results showed that the slr-d7 and slr-d8 variants produced a more extended leaf sheath following GA_3_ treatment, but a reduced length extension compared to WT (Figure 2E).

### 2.3. Altered Transcriptome Profiling in Slr-d7 and Slr-8 Mutants

To understand the impact of dwarfism on gene expression at the whole-genome level, RNA-Seq was conducted to detect transcription profiling changes in WT, slr1-d7 and slr1-d8 lines. RNA-seq results showed that gene expression was altered significantly between WT and the slr1-d7 and slr1-d8 lines (Figure 3A). There are 214 genes upregulated and 154 genes downregulated in the slr1-d7 mutant compared with WT plants. By comparison, 334 genes were upregulated and 104 genes were downregulated in the slr1-d8 line (Figure 3B). Venn diagram analysis revealed 806 genes expressed in both WT and slr1-d7 or slr1-d8 mutants, which may explain the effects of knocking out *SLR1* on plants (Figure 3B,C). Gene ontology (GO) enrichment analysis of the 806 annotated up- and down-regulated genes identified 193 significantly (false discovery rate (FDR < 0.05)) enriched GO terms for the biological process, cellular component, and molecular function categories (Figure 4). Within the biological process category, the enriched differentially expressed genes (DEGs) were mainly associated with the response to the oligopeptide transport (GO:0006857), the intracellular protein transport (GO:0006886), karrikin (GO:0080167), and salt stress (GO:0009651). Within the cellular component category, the enriched DEGs were mainly associated with the plasma membrane (GO:0005886), the membrane (GO:0016020), cytosol (GO:0005829), and the integral component of the membrane (GO:0016021). Within the molecular function category, the DEGs were associated with protein serine/threonine kinase activity (GO:0004674), ATP binding (GO:0005524), and protein binding (GO:0005515) (Appendix A). To confirm the results from the RNA-seq analysis, 38 DEGs in the enriched GO terms were selected in the slr1-d7 and slr1-d8 lines, and their expression levels were confirmed by qRT-PCR analysis. The qRT-PCR results showed that the transcription levels of these genes were consistent with the RNA-seq results (Figure 5). These results indicated that dwarfism of the slr1-d7 and slr1-d8 lines mediates gene expression levels involved in regulating the plant hormone (GA, salicylic acid (SA), jasmonate (JA), IAA, cytokinin (CT) and ethylene (ET)) metabolism, signal transduction and transport (Appendix A).

### 2.4. Key DEGs Related to Biosynthesis and Signaling Pathway of Plant Hormone

In the RNA-seq analysis, the key DEGs related to plant hormone biosynthesis and signal transduction pathways between slr1-d7 vs. WT and slr1-d8 vs. WT were investigated. DEGs between slr1-d7 vs. WT were down-regulated and included the following: gibberellin-regulated protein 2, gibberellin 2-beta-dioxygenase 8 (*GA_2_OX_8_*), E3 ubiquitin-protein ligase (*XERICO*), gibberellin 2-beta-dioxygenase 1 (*GA_2_OX_1_*) in GA biosynthesis, *ERF03*, *ERF110*, *BBM1* in ethylene biosynthesis, and *ILR1* in auxin biosynthesis. Additionally, DEGs between slr1-d7 vs. WT were up-regulated and included the following: gibberellin 2-beta dioxygenase 8, *PIF1*, *PIF4*, *GA_20_OX_2_*, *GAMYB*, *GA_3_OX_2_* in GA biosynthesis, *ERF109*, *ERF39* in ethylene biosynthesis, *LOGL1* in cytokinin biosynthesis, *IAA7* in the auxin biosynthesis pathway, and *JAR1* in jasmonic acid (JA) biosynthesis (Appendix A). Among the DEGs associated with cell plate and leaf morphogenesis, Os05g0432200 and Os04g0407800 seemed to be important for the difference between WT and edited lines. The expression levels of two GA-related genes, especially *GA_20_OX_2_* (gibberellin oxidase) and *GA_3_OX_2_*, were increased in the edited mutant plants compare to WT (Figure 5). The edited mutant lines are required by altering GA responses, at least partially by a defect in the phytohormone signaling process and prevented cell elongation.

## 3. Discussion

CRISPR technology, as a powerful and highly efficient genome editing tool for breeding programs, has been utilized to enable the modification of gene(s) of interest. We report here the CRISPR/Cas9 mutation that potentially could confer a desirable dwarf trait using CRISPR technology. For a long time, plant breeders have used the DELLA gene mutant to reduce plant height [32]. The DELLA protein is one of the main components of the GA signaling pathway and acts as an inhibitor of the GA response. To date, information on dwarfism has shown results in a dominant, semi-dwarf phenotype, such as the observation that the GA signaling repressor DELLA protein or deletion in the N-terminal region suppresses GA signaling [33]. In rice, a total of six slr1-d mutants are known, and these mutants have dark green leaves, reduced internode elongation, and reduced response to GA treatment [28,29,30,32]. Most slr1-d alleles had 1 bp substitutions, resulting in amino acid substitutions in the conserved TVHYNP motif of *SLR1* (Appendix A). In this study, we generated and characterized six new alleles, namely, slr1-d7~slr1-d12, of the dwarf mutants in rice (Appendix A). These mutants showed the same phenotype for leaf color, GA response, and internode elongation with the previously reported slr1-d1~slr1-d6 mutants [28]. Among these mutants, the slr1-d7 gene had a deletion of three nucleotides, resulting in a serine deletion following core sequencing of the TVHYNP motif. Furthermore, the slr1-d8 mutant showed a 1 bp substitution (+T/+T). These two mutants displayed the most obvious and significant mutant phenotypes. The knockout of the slr1-d8 mutant showed a difference in plant height compared to the deletion of the serine residue of the slr1-d7 mutant (Figure 1, Appendix A). This suggests the importance of the TVHYNP domain sequence, as the absence of these amino acids affects the normal metabolism of the SLR1 protein, but the degree of reduction is weaker than previously reported for the slr1-d6 mutant series [28]. We also performed paraffin sections to investigate cell length and cell layer using slr1-d7, slr1-d8 and WT. As a result, it was found that slr1-d7 and slr1-d8 not only showed significantly reduced cell length, but also node thickening, as cell layers increased as compared to WT (Figure 2). In addition, these dwarf mutants showed a decrease in the whole internode length containing a panicle when compared to WT. This result is similar to the characteristics of *dn*-type rice dwarf mutants reported by Takeda [31] and, as a semi-dominant, a decrease in cell length may be a direct cause of shortening clum length in dwarf mutant plants. In shoot elongation tests of dwarf mutant reaction by exogenous GA_3_, the length of the secondary leaf sheath was elongated by GA treatment in WT, but not observed in slr1-d7 and slr1-d8 lines. However, there was a difference between the mutations. These results were similar to those reported in barley *Sln1D* and corn *dwarf 8*, suggesting that a single amino acid deletion or exchange mutation showed an intermediate phenotype depending on plant growth and GAI protein stabilization [13,17,34]. In RNA-seq analysis, the key DEGs related to plant hormone biosynthesis and the signal transduction pathways between slr1-d7 vs. WT and slr1-d8 vs. WT were investigated. DEGs between slr1-d7 vs. WT were down-regulated and included the following: gibberellin-regulated protein 2, gibberellin 2-beta-dioxygenase 8 (*GA_2_OX_8_*), E3 ubiquitin-protein ligase (*XERICO*), gibberellin 2-beta-dioxygenase 1 (*GA_2_OX_1_*) in GA biosynthesis [35], *ERF03*, *ERF110*, *BBM1* in ethylene biosynthesis [36], and *ILR1* in auxin biosynthesis [37]. In RT-PCR and RNA-seq analysis, the expression levels of two GA-related genes, *GA_20_OX_2_* and *GA_3_OX_2_*, increased in the edited mutant line compared to WT, suggesting that these genes convert in the GA_12_ signaling system (Figure 5). The phenomenon of inhibiting cell elongation by altering the GA response due to defects in the signal transduction process of plant hormones was consistent with the results of the *Arabidopsis* mutants [35]. Furthermore, DEGs between slr1-d7 vs. WT were up-regulated and included the following: gibberellin 2-beta dioxygenase 8, *PIF1*, *PIF4*, *GA_20_OX_2_*, *GAMYB*, *GA_3_OX_2_* in GA biosynthesis [38,39,40,41,42,43], *ERF109*, *ERF39* in ethylene biosynthesis [36], *LOGL1* in cytokinin biosynthesis [44], *IAA7* in the auxin biosynthesis pathway [45], and *JAR1* in jasmonic acid (JA) biosynthesis [46]. In summary, our results showed that slr1-d7 and slr1-d8 caused a defect in the phytohormone signaling system process and prevented cell elongation. Furthermore, we suggested that the new slr1-d7~slr1-d12 allelic variants are valuable semi-dominant dwarf alleles with potential application value for molecule breeding using the CRISPR/Cas9 system in rice.

## 4. Materials and Methods 

### 4.1. Plant Materials

Rice variety Dongjin (*Oryza sativar* L., ssp. Japonica) was used for transformation experiments. Plants were grown in GMO greenhouse facilities and rice fields at Hankyong National University in Korea. Harvested seeds were dried to ~14% moisture content and kept in dry conditions at 4 °C.

### 4.2. CRISPR/Cas9 Vector Construction and Rice Transformation

SgRNAs were designed as described in Park et al. [47] to target the TVHYNP motif. The TVHYNPSD amino acid sequence of the *SLR1* gene is encoded by ACCGTGCACTACAACCCCTCGGAC, and the target sequence ACCCCTCGGACCTCTCCCTCCTGG with TGG as the PAM was selected. The 20nt sgRNA scaffold sequence was synthesized by Bioneer co., LTD (Dajeon, Korea). The slr-sgRNA templates were annealed using two primers, 5′-ggcagACCCCTCGGACCTCTCCTCC-3′ and 5′-aaacGGAGGAGAGGTCCGAGGGGTc-3′, and cloned into an *Aar*I-digested *OsU3*:pBOsC binary vector. The Ti-plasmid vector for sgRNA expression, OsU3:*slr1*-sgRNA/pBOsC, and its flanking sequences were confirmed by the Sanger sequencing method and mobilized into *Agrobacterium-tumefaciens* strain EHA105. Transgenic plants were regenerated following a previously described protocol [48]. To confirm the transgene, the independent and transformed lines were analyzed by PCR. Plants derived from tissue culture were rooted and potted into 7 cm pots placed in the glasshouse and gradually acclimatized to the glasshouse conditions.

### 4.3. Targeted Deep Sequencing and Mutation Analysis

Total DNA extraction from plant tissues was performed using the DNA Quick Plant Kit (Inclone, Korea). Targeted deep sequencing analysis was performed following the method described by Jung et al. [46]. All primers used for targeted deep sequencing are listed in Appendix A. Paired-end read sequencing by PCR amplicons was produced with MiniSeq (Illumina, San Diego, CA, USA). All data derived from MiniSeq were analyzed by Cas-Analyzer (http://www.rgenome.net/cas-analyzer), as previously reported by Park et al. [49].

### 4.4. RNA-Seq and Data Analysis

To investigate the transcriptome of edited lines obtained by the *OsSLR1* gene via the CRISPR/Cas9 system, WT, slr1-d7(T/T), and slr1-d8 (−3/−3) plants were used for RNA-seq analysis. Four-week-old leaf tissues were used for RNA extraction, as previously reported by Wang et al. [50]. RNA concentration (A260/A280 and A260/A230) was measured with spectrophotometry (Nanodrop 2000, Thermo Scientific, Hudson, NH, USA). A Bioanalyzer (Agilent Technologies, Santa Clara, CA, USA) was used to evaluate the RNA qualities. Leaf samples of 100 mg were collected from three plants (WT, slr1-d7, and slr1-d8) for RNA-seq analysis. RNA-sequencing was carried out by Macrogen (Seoul, Korea, https://dna.macrogen.com/). Clean reads were produced by removing low-quality reads and mapped to the reference genome (https://plants.ensembl.org/) using TopHat2 (https://ccb.jhu.edu/software/tophat) [51]. Based on location information of the mapped reads, gene expression levels were normalized to reads per kilobase per million mapped reads (RPKM). DEG analyses between the edited plant RNA (slr1-d7, slr1-d8 and WT) were performed using the standard fold change (FC) ≥2 and FDR <0.05. GO analysis was performed as previously reported by Chow et al. [52].

### 4.5. Validation Test of Selected DEGs 

To validate the accuracy of the RNA-sequencing data, qRT-PCR was conducted on twenty-one selected genes. The slr1-d7 and slr1-d8 lines were assessed according to WT, and relative gene expression levels were normalized by the *Actin* gene (XM_015761709). All assays for each gene were performed in triplicate with the same conditions and the RNA-seq data were deposited into the NCBI database.

### 4.6. GA_3_ Treatment

The slr1-d7, slr1-d8, and WT seedlings grown in pots for 4 weeks were sprayed with 50 µM GA_3_ (Sigma-Aldrich, Seoul, Korea). The stock solution of GA_3_ was dissolved in ethanol and added to autoclaved water after cooling to approximately 60 °C to make the final 50 µM solution. The WT plants were treated with water containing equal amounts of ethanol.

### 4.7. Light Microscopy

For the paraffin section, stems and leaves were harvested from slr1-d7, slr1-d8, and WT plants. First, stem tissues were treated with 15% hydrofluoric acid, followed by dehydration with 70% ethanol, removal of it, infiltration, and embedding. For imaging, a 10 μm microtome section was placed on glass slides and floated in a 37 °C water bath containing deionized water. The sections were floated onto clean glass slides and microwaved at 65 °C for 15 min. Following this, the tissues were bound to the glass and each slide was used in chemical staining immediately.

## Figures and Tables

**Figure 1 ijms-21-05492-f001:**
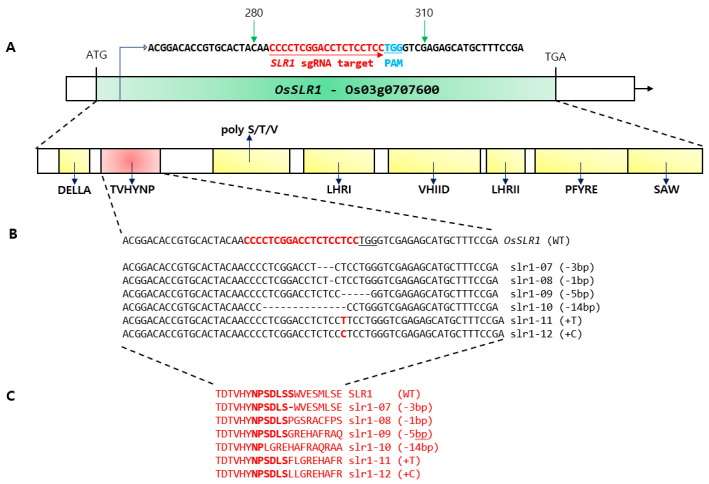
Genome editing in the rice *OsSLR1* gene. (**A**) Design of single guide RNA (sgRNA) sites in the TVHYNP motif; the nuclease cleavage site is represented by the red arrow and the Protospacer Adjacent Motif (PAM) (NGG) appears in blue. DELLA protein organization representing the conserved domains. (**B**) Nucleotide sequence alignment by deep sequence analysis of the sgRNA target region in six mutant lines of transformed rice plants. Deletion and insertion indicated by dash and red letters, respectively. (**C**) Amino acid sequences of the target region in six mutant lines.

**Figure 2 ijms-21-05492-f002:**
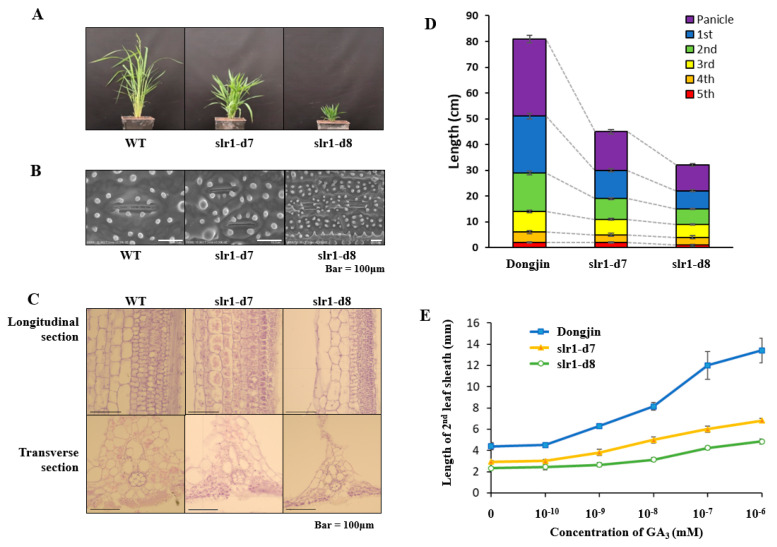
Phenotypic analysis of wild-type (WT) and slr1 mutant plants. (**A**) Phenotype of mature WT and mutant plant lines. (**B**) SEM (scanning electron microscopy) images of rice stomata in slr1-d7, slr1-d8 and WT. (**C**) Longitudinal tissue sections of the main stem at the mature stage in WT and slr1 mutant using paraffin section. Bar: 100 μm. (**D**) Length of internodes in slr1-d7, slr1-d8 and WT. (**E**) Elongation of the second leaf sheath of slr1-d7 and slr1-d8 in response to exogenous treatment with different concentrations of GA_3_. Error bars are SD from the mean (*n* = 3).

**Figure 3 ijms-21-05492-f003:**
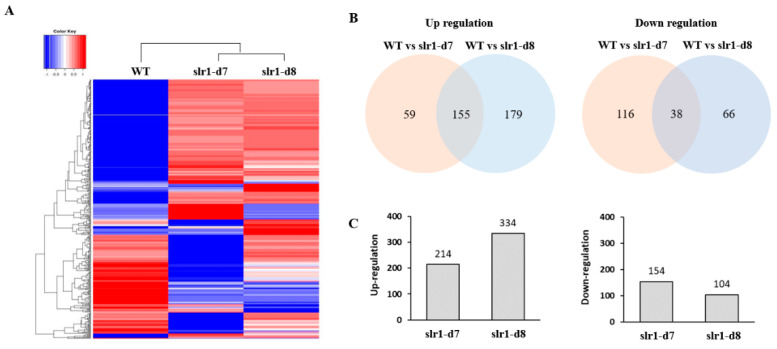
Global gene expression changes in knockout *OsSLR1* in rice. (**A**) Heat map of gene expression between WT vs. slr1-d7 and WT vs. slr1-d8 lines. Red denotes samples with relatively high expression of a given gene and blue denotes samples with relatively low expression. (**B**) Comparison of the number of differentially expressed genes (DEGs) in WT vs. slr1-d7 and WT vs. slr1-d8. (**C**) The number of DEGs up- and down-regulated between WT vs. mutant lines.

**Figure 4 ijms-21-05492-f004:**
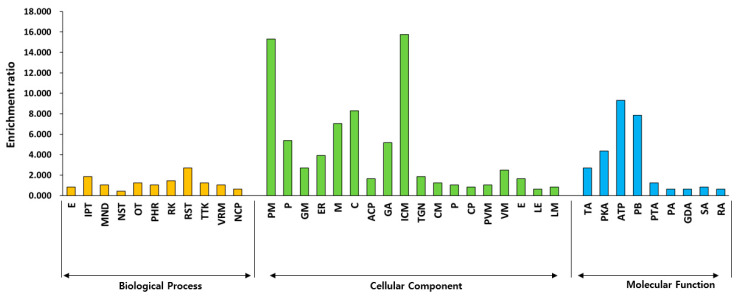
Gene ontology (GO) classification of DEGs shared by WT vs. slr1-d7 and WT vs. slr1-d8 lines. The *x*-axis shows user-selected GO terms, and the *y*-axis shows the enrichment ratio. Biological process: E, exocytosis; IPT, intracellular protein transport; MND, mitotic nuclear division; NST, nitric oxide mediated signal transduction; OT, oligopeptide transport; PHR, plant-type hypersensitive response; RK, response to karrikin; RST, response to salt stress; TTK, transmembrane receptor protein tyrosine kinase signaling pathway; VRM, vegetative to reproductive phase transition of meristem; NCP, nuclear-transcribed mRNA catabolic process. Cellular component: PM, plasma membrane; P, plasmodesma; GM, Golgi membrane; ER, endoplasmic reticulum; M, membrane; C, cytosol; ACP, anchored component of plasma membrane; GA, Golgi apparatus; ICM, integral component of membrane; TGN, trans-Golgi network; CM, chloroplast membrane; P, phragmoplast; CP, cytoplasmic mRNA processing body; PVM, plant-type vacuole membrane; VM, vacuolar membrane; E, endosome; LE, late endosome; EM, endosome membrane. Molecular function: TA, transporter activity; PKA, protein serine/threonine kinase activity; ATP, ATP binding; PB, protein binding; PTA, protein transporter activity; PA, potassium: proton antiporter activity; GDA, glucan endo-1,3-beta-D-glucosidase activity; SA, symporter activity; RA, ribonuclease activity.

**Figure 5 ijms-21-05492-f005:**
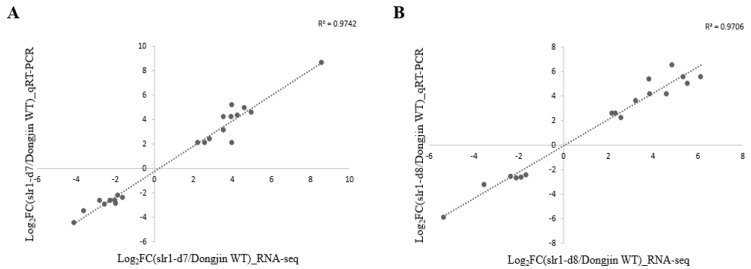
Correlation analysis of gene expression pattern by RNA-Seq and qRT-PCR. (**A**) WT vs. slr1-d7 line, (**B**) WT vs. slr1-d8 line.

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
