# Peer review of "Generation and Transcriptome Profiling of Slr1-d7 and Slr1-d8 Mutant Lines with a New Semi-Dominant Dwarf Allele of SLR1 Using the CRISPR/Cas9 System in Rice"

_ijms, 2020, doi:10.3390/ijms21155492_

Round 1
Reviewer 1 Report
The manuscript titled “Generation and Transcriptome Profiling of slr1-d7 and slr1-d8 Mutant Lines with New Semi-dominant Dwarf Allele of SLR1 using CRISPR/Cas9 System in Rice” by Jung and co-authors describes the generation of a number of novel mutants generated by the CRISPR/cas9 system targeting the specific N-terminal region of the DELLA protein in the crop species rice. The manuscript fundamentally characterise the phenotype and genotype/transcriptome of a selected number (2) of homozygous lines.
While unquestionably interesting the generation and the availability of novel dwarf lines for breeding programs, it fails to bring to the scientific community much else than novel mutants. The manuscripts should be regarded as opposed to a paper as a communication therefore requiring substantial restructuring and critical selection of the data presented.
It is recommended that the authors properly characterise the phenotype of the 6 homozygous mutants by creating a multipannel figure with critical characteristics as opposed to the current only 2 mutants. The authors in the paper acknowledge that the 2 selected mutants (7 and 8) are the most significant, but since the goal of the project was to generate and characterise novel slr1 mutants then, show that data.
General consideration
- Throughout the manuscript, et al., should be italicised
- Throughout the manuscript authors should ensure that numbers and units are separated by a space
- Throughout the manuscript authors must ensure that each reference to mutants and genes all characters are italicised
Major points
- Restructuring and refocussing of the manuscript in a communication with a critical selection of data presented
- Present a clear characterization of the 6 homozygous lines
- Review the manuscript critically for a correct use of grammar and sentence logic.
Major/Minor points
Abstract
Line 18: SLR1 gene …..should be italicised
We generate slr1-d mutants with semi-dominant dwarf to…italicise the gene name and avoid the use of the we and “mutants with semi-dominant dwarf..” should be “mutants with a semi-dominant dwarf phenotype…”
“Line 20-22: delete is a repletion of the above lines
Line 23: “Sixteen genetic edited lines out of 31 transgenic 23 plants were generated with CRISPR/Cas9 system” change to “Sixteen genetic edited lines out of 31 transgenic 23 plants were generated”
Introduction
Line 46: “Phenotypes of…” should be “the phenotypes of…”
Line 62: “..degraded through 62 26S proteasome” should be “degraded through the 26S proteasome..”
Line 65: “…with GA receptor GID1..” should be…” with the GA receptor GID1…”
Results
Line 80: “Editing TVHYNP Domain Encoding OsSLR1 Gene & CRISPR/Cas9 System” should be “80 2.1. Editing of the TVHYNP Domain Encoding the OsSLR1 Gene & CRISPR/Cas9 System”
Lines 80-81 should be improved as it is quite unclear.
Line 83: “…region targeted of OsSLR1 gene..” should be “…region which targeted the OsSLR1 gene..”
Line 86: “All the T0 mutants were dwarf, produce many tillers, and were 87 semi-sterile.” Change to “All the T0 mutants were dwarf, produces many tillers, and were semi-sterile”.
Question to clarify: how was sterility assessed? Were semi in the sense of 50% sterile or else?
Lines 87-90: “Six homozygous mutation types were identified in the edited mutants, a 3-bp deletion that was designated as slr1-d7, and a T-insertion 1-bp deletion that was designated as slr1-d8, a 5-bp deletion named slr1-d9, a 14-bp deletion called slr1-d10, a T-insertion named slr1-d11, and a C90 insertion named slr1-d12” this whole sentence is rather confusing and alternative is proposed below as: “Six homozygous mutants were identified four characterised by few bp deletions and 2 by a few bp insertions. Specifically: a 3-bp deletion and mutant named as slr1-d7, a 1-bp deletion that was designated as slr1-d8, a 5-bp deletion named slr1-d9 and a 14-bp deletion called slr1-d10. The insertion mutants were a T-insertion named slr1-d11, and a C90 insertion named slr1-d12”
Figure 1: appear should be changed to appears
Line 170: “in addition to deducing..” should be : “in addition to reducing..”
Line 110-11 should be improved as the structure is quite cumbersome…
Figure 2: panel an and b are missing errors bars
Panel E: what are the units for the X axe?
Line 125: “Phenotype of mature plant of WT and mutant lines” change to “Phenotype of mature WT and mutant plant lines”
Line 129 “Error bars the SD from the mean (n = 3)” should be “Error bars are SD from the mean (n = 3)”.
Line 148: “..Go terms.” Should be “GO terms..”
Line 151: “..mediates the gene …” should be “mediates gene….”
Figure 3 panel C. indicates what is either up-or-down regulated
Figure 4. low value image remove or else add as a supplementary
Figure 5, interesting correlation but a statement and relative r-values in the test will suffice. Remove
Line 168 “In RNA-seq analysis..” should be “In the RNA-seq analyses..”
Line 170: “…were down-regulated including…..” should be “…were down-regulated included…”
Discussion
Line 182: “CRSPR/Cas9 mutation..” change to “CRISPR/Cas9 mutation..”
Line 184: “…as an inhibitor the GA..” change to “…as an inhibitor of the GA..”
Line 188: “…and reduced respond to GA..” change to “…and reduced response to GA..”
Line 191: “…of the dwarf mutant in rice” change to “…dwarf mutants in rice”
Line 193-194: “The slr1-d7 among these mutants, three-nucleotide deletion (- 194 3/-3) was detected, resulting in a serine deletion following the core sequencing TVHYNP” amend as “Among these mutants the slr1-d7 had a three nucleotides deletion resulting in a serine deletion following core sequencing of the TVHYNP motif”
Line 194: “…mutant was showed 1 bp substitution (+T/+T). These mutants showed the most..” change to ““…mutant showed a 1 bp substitution (+T/+T). These 2 mutants displayed the most..”
Line 200: “…than previously reported slr1-d6.” Change to ““…than previously reported for the slr1-d6 mutant series.”
Line 202: “…only significantly reduced the cell length but also the nodes thickened as the cell layer increased compared to WT” amend as : “only showed significantly reduced cell length but also nodes thickening as cell layers increased as compared to theWT”
Line 203: “…in the all internode” either “..in internodes length” or “..in the whole internode..”
Line 182: “…
Materials and Methods
Line 226-227. Rice varieties should be singular, were should be was. PPlants were grawn in a GMO….
Moreover growth conditions must be specified. Compost type, light conditions, temperature..? etc. etc.
Line 229. A single guide RNAs..delete A
Line 238: “..Transgenic plants were regenerated by as previously described” change to “…Transgenic plants were regenerated as by a previously described protocol []”
Line 239:the inpendent, transformed lines were analyzed. BY? HOW? WHAT? Specify it.
Line 240: “..7 cm pots and placed in the glasshouse. Plants were gradually acclimatized..” change to “..7 cm pots placed in the glasshouse and gradually acclimatized..”
Line 247: “..All data derived from MiniSeq was…” change to ““..All data derived from MiniSeq were…”
Line 248-251. It is a discussion point remove fromM&Ms
Line 254: “..plants WT, slr1-d7(T/T), and slr1-d8 (-3/-3) were..” change to “..WT, slr1-d7(T/T), and slr1-d8 (-3/-3) plants were…”
Line 258. Specified the number of leaves per plants used for the analyses
Line 263: “…DEGs analysis between edited plants (slr1-d7, slr1-d8 and WT were..” change to “…DEGs analyses between edited plant RNA (slr1-d7, slr1-d8 and WT) were..
Line 269. Provide the accession number for the Actin gene
Line 266. Specify the number of genes used for validation
Line 272: “..grown in the pot for 4 weeks” should be “…..grown in pots for 4 weeks..”
The authors refer to media and later to a mix of ethanol and water are those 2 the same? If not specify if yes, choose one and stick to it.
Line 278: “…, followed by dehydration, removal..” of? Subject is amiss here

Author Response
Reviewer 1
General consideration
- Throughout the manuscript, et al., should be italicised
-> Thank you for the critical review. Authors have revised according to the comments.
- Throughout the manuscript authors should ensure that numbers and units are separated by a space
-> Authors have revised according to the comments.
- Throughout the manuscript authors must ensure that each reference to mutants and genes all characters are italicised
-> Authors have revised according to the comments.
Major points
- Restructuring and refocussing of the manuscript in a communication with a critical selection of data presented
-> Authors have revised according to the comments.
- Present a clear characterization of the 6 homozygous lines
-> Authors have revised according to the comments.
- Review the manuscript critically for a correct use of grammar and sentence logic.
-> Authors have revised according to the comments.
Major/Minor points
Abstract
Line 18: SLR1 gene …..should be italicised
We generate slr1-d mutants with semi-dominant dwarf to…italicise the gene name and avoid the use of the we and “mutants with semi-dominant dwarf..” should be “mutants with a semi-dominant dwarf phenotype…”
-> Authors have revised according to the comments.
“Line 20-22: delete is a repletion of the above lines
Line 23: “Sixteen genetic edited lines out of 31 transgenic 23 plants were generated with CRISPR/Cas9 system” change to “Sixteen genetic edited lines out of 31 transgenic 23 plants were generated”
-> Authors have revised according to the comments.
Introduction
Line 46: “Phenotypes of…” should be “the phenotypes of…”
-> Authors have revised according to the comments.
Line 62: “..degraded through 62 26S proteasome” should be “degraded through the 26S proteasome..”
-> Authors have revised according to the comments.
Line 65: “…with GA receptor GID1..” should be…” with the GA receptor GID1…”
-> Authors have revised according to the comments.
Results
Line 80: “Editing TVHYNP Domain Encoding OsSLR1 Gene & CRISPR/Cas9 System” should be “80 2.1. Editing of the TVHYNP Domain Encoding the OsSLR1 Gene & CRISPR/Cas9 System”
-> Authors have revised according to the comments.
Lines 80-81 should be improved as it is quite unclear.
-> Authors have revised according to the comments as follows: “ccording to the structure of the OsSLR1 gene, DELLA and TVHYNP domains are well conserved at the N-terminus.”
Line 83: “…region targeted of OsSLR1 gene..” should be “…region which targeted the OsSLR1 gene..”
-> Authors have revised according to the comments.
Line 86: “All the T0 mutants were dwarf, produce many tillers, and were 87 semi-sterile.”
Change to “All the T0 mutants were dwarf, produces many tillers, and were semi-sterile”.
Question to clarify: how was sterility assessed? Were semi in the sense of 50% sterile or else?
-> Authors have revised according to the comments.
Lines 87-90: “Six homozygous mutation types were identified in the edited mutants, a 3-bp deletion that was designated as slr1-d7, and a T-insertion 1-bp deletion that was designated as slr1-d8, a 5-bp deletion named slr1-d9, a 14-bp deletion called slr1-d10, a T-insertion named slr1-d11, and a C90 insertion named slr1-d12” this whole sentence is rather confusing and alternative is proposed below as: “Six homozygous mutants were identified four characterised by few bp deletions and 2 by a few bp insertions. Specifically: a 3-bp deletion and mutant named as slr1-d7, a 1-bp deletion that was designated as slr1-d8, a 5-bp deletion named slr1-d9 and a 14-bp deletion called slr1-d10. The insertion mutants were a T-insertion named slr1-d11, and a C90 insertion named slr1-d12”
-> Authors have revised according to the comments.
Figure 1: appear should be changed to appears
-> Authors have changed according to the comments.
Line 170: “in addition to deducing..” should be : “in addition to reducing..”
-> Authors have revised according to the comments.
Line 110-11 should be improved as the structure is quite cumbersome…
-> Authors have revised according to the comments.
Figure 2: panel an and b are missing errors bars
-> Authors have included error bars according to the comments.
Panel E: what are the units for the X axe?
-> Authors have included the unit according to the comments.
Line 125: “Phenotype of mature plant of WT and mutant lines” change to “Phenotype of mature WT and mutant plant lines”
-> Authors have revised according to the comments.
Line 129 “Error bars the SD from the mean (n = 3)” should be “Error bars are SD from the mean (n = 3)”.
-> Authors have revised according to the comments.
Line 148: “..Go terms.” Should be “GO terms..”
-> Authors have revised according to the comments.
Line 151: “..mediates the gene …” should be “mediates gene….”
-> Authors have revised according to the comments.
Figure 3 panel C. indicates what is either up-or-down regulated
-> Authors have revised according to the comments.
Figure 4. low value image remove or else add as a supplementary
-> Authors have revised according to the comments.
Figure 5, interesting correlation but a statement and relative r-values in the test will suffice. Remove
-> We think that the correlation graph need to be included for readers to get understood easily.
Line 168 “In RNA-seq analysis..” should be “In the RNA-seq analyses..”
-> Authors have revised according to the comments.
Line 170: “…were down-regulated including…..” should be “…were down-regulated included…”
-> Authors have revised according to the comments.
Discussion
Line 182: “CRSPR/Cas9 mutation..” change to “CRISPR/Cas9 mutation..”
-> Authors have revised according to the comments.
Line 184: “…as an inhibitor the GA..” change to “…as an inhibitor of the GA..”
-> Authors have revised according to the comments.
Line 188: “…and reduced respond to GA..” change to “…and reduced response to GA..”
-> Authors have revised according to the comments.
Line 191: “…of the dwarf mutant in rice” change to “…dwarf mutants in rice”
-> Authors have revised according to the comments.
Line 193-194: “The slr1-d7 among these mutants, three-nucleotide deletion (- 194 3/-3) was detected, resulting in a serine deletion following the core sequencing TVHYNP” amend as “Among these mutants the slr1-d7 had a three nucleotides deletion resulting in a serine deletion following core sequencing of the TVHYNP motif”
-> Authors have revised according to the comments.
Line 194: “…mutant was showed 1 bp substitution (+T/+T). These mutants showed the most..” change to ““…mutant showed a 1 bp substitution (+T/+T). These 2 mutants displayed the most..”
-> Authors have revised according to the comments.
Line 200: “…than previously reported slr1-d6.” Change to ““…than previously reported for the slr1-d6 mutant series.”
-> Authors have revised according to the comments.
Line 202: “…only significantly reduced the cell length but also the nodes thickened as the cell layer increased compared to WT” amend as : “only showed significantly reduced cell length but also nodes thickening as cell layers increased as compared to theWT”
-> Authors have revised according to the comments.
Line 203: “…in the all internode” either “..in internodes length” or “..in the whole internode..”
-> Authors have revised according to the comments.
Line 182: “…
-> Authors have revised according to the comments.
Materials and Methods
Line 226-227. Rice varieties should be singular, were should be was. PPlants were grawn in a GMO….
Moreover growth conditions must be specified. Compost type, light conditions, temperature..? etc. etc.
-> Authors have revised according to the comments as follows: “Harvested seeds were dried to ~ 14% moisture content and kept in dry conditions at 4 °C.”
Line 229. A single guide RNAs..delete A
-> Authors have revised according to the comments.
Line 238: “..Transgenic plants were regenerated by as previously described” change to “…Transgenic plants were regenerated as by a previously described protocol []”
-> Authors have revised according to the comments.
Line 239:the inpendent, transformed lines were analyzed. BY? HOW? WHAT? Specify it.
-> Authors have revised according to the comments as follows: “To confirm the transgene the independent, transformed lines were analyzed by PCR.”
Line 240: “..7 cm pots and placed in the glasshouse. Plants were gradually acclimatized..” change to “..7 cm pots placed in the glasshouse and gradually acclimatized..”
-> Authors have revised according to the comments.
Line 247: “..All data derived from MiniSeq was…” change to ““..All data derived from MiniSeq were…”
-> Authors have revised according to the comments.
Line 248-251. It is a discussion point remove fromM&Ms
-> Authors have revised according to the comments.
Line 254: “..plants WT, slr1-d7(T/T), and slr1-d8 (-3/-3) were..” change to “..WT, slr1-d7(T/T), and slr1-d8 (-3/-3) plants were…”
-> Authors have revised according to the comments.
Line 258. Specified the number of leaves per plants used for the analyses
-> Authors have revised according to the comments.
Line 263: “…DEGs analysis between edited plants (slr1-d7, slr1-d8 and WT were..” change to “…DEGs analyses between edited plant RNA (slr1-d7, slr1-d8 and WT) were..
-> Authors have revised according to the comments.
Line 269. Provide the accession number for the Actin gene
-> Authors have revised according to the comments as follows: “Actin gene (XM_015761709)”
Line 266. Specify the number of genes used for validation
-> Authors have revised according to the comments.
Line 272: “..grown in the pot for 4 weeks” should be “…..grown in pots for 4 weeks..”
The authors refer to media and later to a mix of ethanol and water are those 2 the same? If not specify if yes, choose one and stick to it.
-> Authors have revised according to the comments.
Line 278: “…, followed by dehydration, removal..” of? Subject is amiss here
-> Authors have revised according to the comments as follows: “followed by dehydration by 70% ethanol, removal of it,”

Reviewer 2 Report
The manuscript describes the generation of six new mutants (slr1-d7 to slr1-d12) in rice SLR1 gene. These mutants showed the same phenotype as previously reported ones on rice (slr1-d1 to slr-d6). In the manuscript, the mutants slr1-d7 and slr1-d8 have been characterized, and transcriptomic analysis performed.
I consider that the authors should complete the transcriptomic analysis in the manuscript.
- In the abstract the authors mentioned the involvement of GA20OX2 and GA3OX2, among the different DEG found. If the authors highlighted this in abstract, the data should be shown on the results section (perhaps with the qRT-PCR data).
- I cannot not find any information regarding the qRT-PCR on material and methods or as supplementary.
- The discussion about the RNAseq experiment should be more intense.
Author Response
Reviewer 2
The manuscript describes the generation of six new mutants (slr1-d7 to slr1-d12) in rice SLR1 gene. These mutants showed the same phenotype as previously reported ones on rice (slr1-d1 to slr-d6). In the manuscript, the mutants slr1-d7 and slr1-d8 have been characterized, and transcriptomic analysis performed.
I consider that the authors should complete the transcriptomic analysis in the manuscript.
- In the abstract the authors mentioned the involvement of GA20OX2 and GA3OX2, among the different DEG found. If the authors highlighted this in abstract, the data should be shown on the results section (perhaps with the qRT-PCR data).
--> Authors have included the explanation of GA20OX2 and GA3OX2 in Results and Discussion
- I cannot not find any information regarding the qRT-PCR on material and methods or as supplementary.
--> Authors discribed qRT-PCR process on Material and Methods.
- The discussion about the RNAseq experiment should be more intense.
--> We discribed RNA-seq in Discussion.

Round 2
Reviewer 1 Report
The manuscript was improved upon suggestion. Still, review carefully for the accurate use of italics for the mutants names as in places are not italicised.
Author Response
Reviewer 1
The manuscript was improved upon suggestion. Still, review carefully for the accurate use of italics for the mutant’s names as in places are not italicised.
-> Thank you for the critical review. Authors have revised according to the comments.
Reviewer 2 Report
The authors have inserted the same paragraph in Results (lines 189-193) and in Discussion (lines 232-236). They should differentiate between these two parts and state a paragraph regarding the result and a different one regarding its discussion.
Author Response
Reviewer 2
The authors have inserted the same paragraph in Results (lines 189-193) and in Discussion (lines 232-236). They should differentiate between these two parts and state a paragraph regarding the result and a different one regarding its discussion.
-> Thank you for the critical review. Authors have revised according to the comments.
Results (lines 189-193): The expression levels of two GA-related genes, especially, GA20OX2 (Gibberellin oxidase) and GA3OX2, were increased in the edited mutant plants compare to WT (Figure 5). The edited mutant lines are required by altering GA responses, at least partially by a defect in the phytohormone signaling process and prevented cell elongation.
Discussion (lines 232-236): In RT-PCR and RNA-seq analysis, the expression levels of two GA-related genes, GA20OX2 and GA3OX2, increased in the edited mutant line compared to WT suggests that these genes act to convert in the GA12 signaling system (Figure 5). The phenomenon of inhibiting cell elongation by altering the GA response due to defects in the signal transduction process of plant hormones was consistent with the results of the Arabidopsis mutants [36].